# Rapid and Accurate Detection of *Gnomoniopsis smithogilvyi* the Causal Agent of Chestnut Rot, through an Internally Controlled Multiplex PCR Assay

**DOI:** 10.3390/pathogens11080907

**Published:** 2022-08-12

**Authors:** Matias Silva-Campos, Pavani Nadiminti, David Cahill

**Affiliations:** 1School of Life and Environmental Sciences, Geelong Waurn Ponds Campus, Deakin University, Geelong, VIC 3216, Australia; 2Department of Animal, Plant and Soil Sciences, School of Agriculture, Biomedicine and Environment, La Trobe Institute for Agriculture and Food, AgriBio, La Trobe University, Melbourne, VIC 3086, Australia

**Keywords:** *Gnomoniopsis smithogilvyi*, multiplex PCR, internal control, virulence, chestnut rot, *Castanea sativa*

## Abstract

The fungus *Gnomoniopsis smithogilvyi* is a significant threat to the production of sweet chestnut (*Castanea sativa*) nuts in Australia and worldwide. The pathogen causes nut rot, which leads to substantial production losses. Early and accurate diagnosis of the disease is essential to delineate and implement control strategies. A specific and sensitive multiplex PCR was developed based on the amplification of three barcode sequences of *G. smithogilvyi*. The assay reliability was enhanced by including the amplification of a host gene as an internal control. Primers were thoroughly evaluated in silico before assessing them in vitro. Primer annealing temperature and concentration were optimised to enhance the assay sensitivity and specificity. The assay detection limit ranged between 0.1 and 1.0 pg (5 and 50 fg/μL) of genomic DNA per reaction. No cross-reactivity was observed with genomic DNA from closely and distantly related fungal species. We also characterised Australian *G. smithogilvyi* isolates phenotypically and genotypically and found significant differences in morphologic and virulence traits of the isolates. An understanding of the virulence of *G. smithogilvyi* and the availability of a reliable and accurate diagnostic technique will enable earlier detection of the pathogen, which will contribute to effective control strategies for the disease.

## 1. Introduction

The sweet chestnut (*Castanea sativa*) is a historically important fruiting tree native to southern Europe [1]. Likely due to the migration of Europeans, the sweet chestnut was introduced to the Americas, Oceania, and Southern and Western Asia [2,3]. In Australia, the early cultivation of this species in the 1850s provided chestnuts to farmer households and local markets [4]. Currently, the Australian chestnut industry sustains approximately 300 small-scale growers and is becoming an emerging and profitable niche market [5]. For example, the production value in Australia increased by 19.2% between 2019 and 2021 [6]. However, the success of this market and, thus, grower livelihoods are threatened by a significant phytosanitary problem, chestnut rot.

The fungus *Gnomoniopsis smithogilvyi* (syn. *Gnomoniopsis castaneae*) (Gnomoniaceae, Diaporthales) is considered the main causal agent of chestnut rot in Australia [7,8,9]. The fungus has both endophytic and pathogenic lifestyles [10]. Under the appropriate conditions, *G. smithogilvyi* becomes pathogenic, infecting nuts early in the season and turning the solid nut endosperm into a chalky and soft tissue [11] of no marketable value. It is not only the Australian chestnut industry that is threatened by this disease, as *G. smithogilvyi* has also been reported to cause nut rot in Chile [12], Greece [13], India [14], Italy [9], Portugal [15], Spain [16] and Switzerland [17], among other countries. Peaks of disease can fluctuate yearly, reaching levels of significant magnitude in some seasons. For example, Dennert et al. [17] found infections of mature nuts in Switzerland of up to 91%. In Australia, Shuttleworth et al. [7] also reported up to 72% of rotted nuts to be infected by *G. smithogilvyi*. Thus, this pathogen is a severe risk to the chestnut industry and warrants the implementation of a comprehensive control strategy.

Implementing control strategies, such as the use of fungicides [18], for plant diseases is time-dependent and relies on the detection and identification of the causative agent precisely and rapidly [19,20]. Currently, numerous plant pathogens, including ascomycetes [21], basidiomycetes [22,23] and oomycetes [24], are detected quickly and accurately using PCR-based techniques [25]. However, PCR assays are susceptible to failure due to multiple factors, including the presence of PCR inhibitors, faulty reagents, or human error [26], which might lead to a false-negative result. To minimise this risk, a reliable diagnostic technique should include internal controls [27]. Currently, for the detection of *G. smithogilvyi*, only a monoplex PCR, real-time PCR and loop-mediated isothermal amplification (LAMP) assay have been reported [28,29,30]. However, the results and reliability of such techniques come under question when reaction internal controls are not included [31,32]. Therefore, a primary goal of this study was to develop an internally controlled, reliable, and sensitive diagnostic assay. The widely used fungal barcodes—Internal Transcribed Spacer region (ITS) [33], *translational elongation factor 1α* (*TEF*) and *β-tubulin* (*TUB*) [34]—were combined in a multiplex PCR (mPCR) for the specific detection of *G. smithogilvyi*.

The implementation of disease management strategies also requires an understanding of the variation in virulence among isolates of a pathogen, for example, for the selection of resistant crop cultivars [35]. Pathogen aggressiveness or virulence can be defined quantitatively and is measured as the degree of damage caused by the pathogen to the host tissue [36,37]. In the context of chestnut rot, little is known about the virulence profile of *G. smithogilvyi* morphotypes. In Australia, only very limited studies have examined the differences among pathogen isolates [10]. Therefore, to complement our diagnostic assay development, we also carried out a comprehensive characterisation of Australian *G. smithogilvyi* morphotypes, which included a morphological, phylogenetic and comparative aggressiveness analysis.

## 2. Results

### 2.1. Isolation and Identification of G. smithogilvyi

Nuts and burrs with symptoms of infection were sourced from four locations in Victoria, Australia (Figure 1). We isolated twenty-four isolates in pure cultures that were morphologically identified as *G. smithogilvyi-like*. Isolates were further confirmed to the species level by amplifying and sequencing the ITS region. All isolates had 100% identity to the ex-type culture CBS 130190 (ID no. NR_166040). In addition, we also recovered from the samples colonies that resembled other fungal taxa. One representative of each taxon, ten in total, was subcultured to obtain axenic cultures. We extracted DNA from the colonies and carried out amplification of the ITS region to identify the species. The results from the NCBI-Blast search showed that the fungal specimen sequences were 99% or more identical to the species *Alternaria alternata* (MK979373), *Aspergillus* sp. (KJ567462), *Cladosporium* sp. (KU238142), *Clonostachys rosea* (MT448899), *Epicoccum nigrum* (MT548679), *Fusarium ramigenum* (MH862657), *Mucor* sp. (MT777429), *Nigrospora oryzae* (HM999906), *Penicillium commune* (MN105270), and *Phoma fungicola* (KF293763).

### 2.2. Phylogenetic Analysis of G. smithogilvyi

A total of fifty-five ITS1-5.8S-ITS2 sequences of the chestnut pathogen *G. smithogilvyi* were used for phylogenetic analysis, including 28 sequences from different countries retrieved from GenBank (Figure 2). These countries included Chile, Greece, India, Italy, Portugal, Spain, and Switzerland. The constructed maximum likelihood (ML) consensus tree was derived from 1000 bootstrap replications having a log-likelihood of −3084.25. The phylogenetic tree clustered all Australian and overseas *G. smithogilvyi* isolates into one well-supported monophyletic clade (bootstrap = 99), thus corroborating the identity of the 24 isolates tested in this study. Moreover, the analysis based on ITS sequences confirmed the phylogenetic divergence between the *G. smithogilvyi* isolates and the closely related species *G. chinensis*, *G. comari*, *G. daii*, *G. fructicola*, and *G. idaeicola*.

The comparison of the ITS sequence alignment properties and genetic variability within *G. smithogilvyi* isolates is presented in Table 1. Isolates from Switzerland showed the shortest sequence length with a mean of 530 bp, whereas isolates from Chile had the longest nucleotide sequences with an average of 611 bp. On the other hand, the average sequence length for Australian isolates was 568 bp. In terms of site variability, the percentage of conserved sites varied from 76.04% in the group from Spain to 90.47% in isolates from Chile. The Australian representatives showed 84.22% of conserved sites. On the other hand, the lowest percentage (0%) of variable sites was found in the isolates from Greece, Portugal and Switzerland, whereas isolates from India showed the highest percentage (4.01%). The Australian group showed 3.12% of variable sites.

The Kimura 2-parameter model showed that the intrapopulation genetic divergence was lower than 1% among all the groups analysed. The lowest percentage (0.00% ± 0.00) of intrapopulation divergence was found among isolates from Chile, Greece, Italy, Portugal, and Switzerland, whereas the isolates from India had the highest percentage (0.78% ± 0.23). Australian representatives showed 0.02% ± 0.02 of intrapopulation divergence. On the other hand, the analysis of interpopulation divergence showed null nucleotide variation between Switzerland, Chile, Greece, and Portugal isolates. On the contrary, the Italian isolates were the most divergent group compared with the other populations. The maximum interpopulation variation (1.08% ± 0.39) was observed between the isolates from Italy and India.

### 2.3. Species-Specific Primer Design

We derived and analysed ten primer pairs for each *G. smithogilvyi* gene in silico, selecting only the most optimal primer sequences (Table 2). These sequences were 20 or 21 nucleotides long and were 100% identical to the respective *G. smithogilvyi* sequences; translation elongation factor 1α (*TEF*); Internal Transcribed Spacer (ITS), and β-tubulin (*TUB*). Moreover, the selected nucleotide sequences were within the recommended GC% content range (45–60%), with the primer GsEFA-F having the minimum percentage (45%) and primer GsITS-F the maximum (57%). Melting temperature (T_m_) varied from 52.1 to 58 °C. The evaluation of the primer sequences in all possible combinations showed a low potential for forming dimers or other secondary structures. None of the combinations had a Gibbs free energy (ΔG kcal/mol) lower than the established threshold of −9 kcal/mol. During the design process, special attention was given to the theoretical size of amplicons. To facilitate electrophoretic separation and visualisation of the agarose gel, we selected only primers that yielded amplicons with substantial size differences. All amplicons were 100 bp different or more, and the minimum difference was 116 bp between amplicons *TEF* and ITS.

### 2.4. Multiplex PCR Optimisation

#### 2.4.1. Annealing Temperature

Optimisation of the annealing temperature (T_a_) was carried out through a thermal gradient assay from 51.8 to 59.7 °C (Figure 3). The results showed that all primer pairs effectively amplified the intended sequences. The resulting PCR products were separated by electrophoresis in 1% agarose. The amplicons derived from the amplification with the specific primers varied in terms of size; GsEFA/*TEF* = 483 bp, GsITS/ITS = 367 bp, GsTUB/*TUB* = 203 bp, and Cmcs1/*petD* = 109 bp. However, we observed unintended amplification in the lower range of the temperatures tested. For example, primer pair GsEFA amplified untargeted regions of *G. smithogilvyi* DNA between 51.8 and 55.6 °C.

This optimisation assay also revealed the effect of the temperature on the intensity of the PCR product *petD*. The intensity of this amplicon was considerably reduced at T_a_ higher than 58.2 °C, suggesting that the efficiency of annealing of the primer pair Cmcs1 is reduced at high temperatures. On the contrary, the primers GsITS and GsTUB were consistently stable across all temperatures tested, as shown by the uniform intensity of the PCR products produced and the lack of unintended amplicons. Based on these results, the optimal T_a_ for mPCR was established at 56.9 °C.

#### 2.4.2. Primer Concentration

We evaluated the primer concentration in different combinations at the optimised T_a_ of 56.9 °C. We initially used 0.25 μM of each primer in a single reaction to then reduce the concentration of only one primer pair at a time at intervals of 0.5 μM (Figure 4). The results showed that the primer concentration was a limiting factor for obtaining uniform and intense amplicon bands. We found that the intensity of the PCR product *TEF* is reduced considerably at concentrations of GsEFA ≤ 0.15 μM. Concentrations of GsITS ≤ 0.10 μM reduced the strength of amplicon *ITS*, and GsTUB ≤ 0.10 μM affected *TUB* amplicon intensity. Similarly, concentrations of the internal control primers Cmcs1 ≤ 0.20 μM limited the amplification of host gene *petD*. Based on the results, we determined that performing an efficient and optimal mPCR requires a concentration of 0.25 μM for primers GsEFA and Cmcs1, and 0.15 μM for primers GsITS and GsTUB.

### 2.5. Multiplex PCR Specificity

The specificity of the designed primers was confirmed by carrying out the mPCR assay in vitro at the optimised conditions of T_a_ and primer concentration (Figure 5). The mPCR assay showed to be highly specific for *G. smithogilvyi*, amplifying effectively 100% of the target sequences of the twenty-four isolates. In concordance, only PCR products at the expected sizes were produced. Moreover, cross-reactivity or formation of artefacts was not observed when using gDNA from the two closely related species *G. fructicola* and *G. idaeicola*. Similarly, amplification of unintended sequences was not produced with any of the ten fungal species found coexisting with *G. smithogilvyi* in chestnuts. Furthermore, the amplification of the internal control gene *petD* in each reaction confirmed that the negative results are due to the absence of *G. smithogilvyi* gDNA and not due to a reaction failure.

### 2.6. Multiplex PCR Detection Limit

The detection limit of the developed mPCR was determined through a dilution series with concentrations ranging from 1000 pg down to 0.1 pg of *G. smithogilvyi* gDNA in a final volume of 20 μL (Figure 6). In addition, each reaction tube was spiked with 1 μL (10 ng/μL) of *C. castanea* gDNA to mock the high level of the host DNA present in fresh chestnut samples. As expected, the intensity of the amplified bands decreased gradually as a result of the reduction in total gDNA. On the other hand, the amplification of the internal control gene *petD* by the primer pair Cmcs1 showed consistent band intensity across all concentrations evaluated. This suggests that even at high concentrations of host background DNA, the assay is capable of detecting *G. smithogilvyi* gDNA. Based on the electrophoretic separation of the mPCR products, the detection limit of the assay was situated between 0.1 and 1 pg of gDNA in 20 μL. This range is equal to 5–50 fg/μL when expressed as mass/volume. Of note was that at 0.1 pg, only the primer pair GsTUB, for the *TUB* gene, yielded an amplicon, although the resulting band was faint.

### 2.7. Multiplex PCR Validation

The newly developed mPCR assay was validated with 50 nuts showing varying symptoms (Figure 7 and Appendix A). Symptomless nuts (*n* = 31) had bright yellowish endosperms. In contrast, symptomatic nuts (*n* = 19) showed varied infection levels, from slightly decorated spots to completely chalky endosperms. In total, 27 (54%) nuts were negative, and 23 (46%) nuts were positive for *G. smithogilvyi*. The analysis of infection levels per group showed that in symptomless nuts, 8 (16%) nuts were positive, and 23 (46%) nuts were negative for the pathogen. On the other hand, as expected, infection by the pathogen was significantly higher (*p* < 0.0001) in symptomatic than symptomless nuts. A total of 15 symptomatic nuts (30%) were found to be positive for *G. smithogilvyi*, and 4 (8%) nuts were negative. Of note is that the cultures derived from the four symptomatic nuts that were mPCR negative to *G. smithogilvyi* yielded fungal colonies that were morphologically identical to previously isolated *Alternaria* and *Mucor* species.

### 2.8. Morphological Characterisation of Colony and Conidia of Isolates

The axenic cultures of the anamorphic *G. smithogilvyi* isolates were grown on PDA at 25 °C for 6 days before measuring radial growth and recording micromorphological characters (Figure 8). *Culture characteristics*: Colony circular, margin mostly undulate and rarely entire. Margin white-ish to grey-ish, brown-ish towards the centre, with the reverse of colonies, pale yellow (Figure 8A–H). Mycelium raised and woolly, rarely flat, generally displaying a concentric growth pattern. Conidiomata abundant after ten days of incubation, yellowish to black, irregularly distributed, globose when raised, rugose when immersed into the medium (Figure 8I–L). Isolates were induced to sporulate on CMA at room temperature for 10 days before comparing conidial morphology and size. This incubation period showed to be enough for all isolated colonies to reach the edge of plates (90 mm) and produce conidia profusely. *Conidium characteristics*: hyaline, aseptate, often biguttulate, mainly straight ovoid to ellipsoid, occasionally curved, allantoid or pyriform (Figure 8M–P).

The analysis of colony growth and conidial size revealed significant differences among populations of *G. smithogilvyi* (Figure 9). The assessment of colony diameter showed that the mean growth of isolates from Stanley (70.17 mm ± 1.33) was the highest recorded across the four populations (Figure 9A). The mean growth of this group was statistically different (*p* < 0.01) from that of isolates from Bright (64.79 mm ± 1.34), which had the lowest growth mean of the four populations analysed. However, the mean of the Stanley population did not differ significantly (*p* > 0.05) from the mean growth of the populations from Fumina (68.53 mm ± 0.78) and Wandiligong (66.13 mm ± 0.84). The analysis at the level of individual isolates showed that both groups, Stanley and Bright, contained the two slowest and fastest-growing morphotypes of all isolates studied. In Stanley, STA1 (53.63 mm ± 0.68) was the slowest, whereas the STA2 (81.63 mm ± 0.37) was the fastest-growing morphotype. On the other hand, in Bright, the isolates BRI5 (54.25 mm ± 0.62) and BRI3 (80.38 mm ± 0.82) were the slowest and fastest morphotypes, respectively (Appendix AA).

The analysis of conidial size showed that the mean length of morphotypes from Stanley (6.03 μm ± 0.04) was significantly greater (*p* < 0.05) than those of Wandiligong (5.86 μm ± 0.04), (*p* < 0.01) Bright (5.83 μm ± 0.04) and (*p* < 0.001) Fumina (5.79 μm ± 0.04) (Figure 9B). On the other hand, the mean width of isolates from Stanley (2.40 μm ± 0.02) and Bright (2.49 μm ± 0.02) was not statistically different (*p* > 0.05). However, the conidia from Stanley were significantly wider (*p* < 0.01) than Fumina (2.28 μm ± 0.03) and Bright was significantly wider (*p* < 0.001) than Wandiligong (2.35 μm ± 0.02) and (*p* < 0.0001) Fumina. Conidia from Fumina had the smallest mean width of all four groups studied (Figure 9C). The mean Q (length/width) ratio described the morphotypes from Fumina as the most elongated group (Q = 2.61) and isolates from Bright as the less elongated morphotypes (Q = 2.39) (Figure 9D). Furthermore, the analysis of length and width at the individual morphotype level showed that STA6 was the longest (6.35 μm ± 0.08) and widest isolate (2.69 μm ± 0.04), whereas FUM4 was the shortest (5.19 μm ± 0.09) and smallest isolate (2.01 μm ± 0.08) of all morphotypes analysed (Appendix AB–D).

### 2.9. Assessment of G. smithogilvyi Isolate Virulence In Vitro

The virulence of *G. smithogilvyi* isolates was evaluated in vitro in wounded mature nuts that were inoculated with three representative morphotypes from each population. Wounded nuts inoculated with sterile agar plugs were used as control treatments. The lesions caused by the pathogen in the endosperms were measured after 8 days post-inoculation at 25 °C (Figure 10). Controls showed no lesion development. Visually, the infections were localised at the point of inoculation and consisted of lesions surrounded by a dark ring. Degradation of nut endosperm was evident in the area of infection and distinguishable from the uninfected areas of the nut (Figure 10A–D). The evaluation of lesion sizes showed that the morphotypes from Stanley were the most virulent. The mean lesion area of this group was 0.78 ± 0.04 cm^2^, which was significantly higher (*p* < 0.01) to that of Wandiligong (0.56 ± 0.04 cm^2^), Bright (0.57 ± 0.04 cm^2^) and Fumina (0.60 ± 0.04 cm^2^) (Figure 10E). The mean lesion of the last three populations did not differ significantly (*p* > 0,05). The analysis at the level of individual isolates showed that STA6 was the most virulent morphotype, causing lesions of 0.80 ± 0.07 cm^2^. In contrast, WAN6 was the least virulent isolate causing lesions of 0.45 ± 0.08 cm^2^ (Appendix A).

## 3. Discussion

*Gnomoniopsis smithogilvyi* is considered to be the principal causal agent of chestnut rot, putting at risk chestnut production in Australia and worldwide. In this study, we developed and validated a reliable, sensitive and specific multiplex PCR technique for detecting *G. smithogilvyi* in nuts. We also characterised phenotypically and phylogenetically analysed isolates of the pathogen sourced from the main chestnut-producing areas in Australia.

Molecular techniques such as PCR are cornerstone tools in plant pathology. They allow the early detection, accurate diagnosis and delivery of timely information for the implementation of effective control strategies. However, assuring the quality of these techniques is of paramount importance to provide confidence in the results. Recent studies showed that the use of PCR internal controls is crucial for the minimisation of false-negative results, thus enhancing the reliability of this type of technique [31,38]. Endogenous genes, such as those of the host, are suggested as suitable candidates for the internal control. The successful amplification of these genes shows that the extraction of DNA, its integrity and the level of PCR inhibitors, factors that are known to cause false-negative results in PCR-based assays [39], are within optimal parameters [40]. In this study, we enhanced the reliability of the newly developed mPCR assay by including primers for the amplification of the *C. sativa* gene *petD* as the internal control. This is the first technique developed for detecting *G. smithogilvyi* that includes the use of an internal control.

The accuracy of PCR detection techniques depends strongly on the specificity of the designed primers. Hence, a thorough analysis in silico should be carried out to select the most promising primers before evaluating them in vitro. This analysis should consider optimal primer sequence length, G-C content, and melting temperature, as these are essential parameters to ensure an efficient and specific amplification of the intended genes [41]. Additionally, only primer sequences with low potential for forming dimers or other secondary structures should be selected, as these might impact negatively on their specificity [42,43]. In this study, we designed and selected only primers within the optimal parameters of sequence length, G-C content, melting temperature and low likelihood of producing secondary structures. On the other hand, primer specificity should also be evaluated in vitro using a number of the target species isolates, closely related species and also with species that inhabit the host with the pathogen [27].

Our phylogenetic analysis showed that the twenty-four *G. smithogilvyi* isolates obtained from different locations were a well-conserved group with low genetic variation. The use of the developed mPCR with this extensive panel of isolates demonstrated the high accuracy of the developed mPCR, as all the intended genes from the pathogen were effectively amplified. Moreover, the low genetic divergence observed between the Australian and overseas populations suggests that our mPCR has the potential for detection of *G. smithogilvyi* morphotypes from other chestnut-growing countries. Additionally, the evaluation in vitro of the mPCR specificity showed no cross-reactivity with gDNA from the closely related *Gnomoniopsis* species and the other ten distantly related fungal taxa that were also found colonising chestnut tissues. The diagnostic assay developed here is thus highly specific for *G. smithogilvyi*.

Optimisation of the annealing temperature and primer working concentrations enhance the detection limit of PCR techniques [43]. Our experiments showed that the optimal T_a_ was 56.9 °C and that the primer pair concentration ranged from 0.15 to 0.25 μM. Cross amplification or formation of secondary structures was not observed during the determination of the optimal primer concentration. Under the optimised parameters, the detection limit of our endpoint mPCR ranged from 0.1 to 1 pg (5–50 fg/μL) of *G. smithogilvyi* gDNA per reaction. Our mPCR assay requires a similar amount of time to perform and similar instrumentation, but it is more sensitive than previously reported endpoint mPCR-based techniques for other fungal species. For example, Iturralde Martinez et al. [44] developed an mPCR for the detection of three *Ophiosphaerella* species. However, the technique was unable to detect the species simultaneously at concentrations of gDNA lower than 10 ng/μL and lacked an internal control. Compared to other endpoint PCR-based techniques, our mPCR is also more sensitive and reliable. For example, Chaisiri et al. [45] developed a monoplex PCR for detecting the fungal pathogen *Diaporthe citri* in citrus for which the detection limit was 10 pg. For chestnuts, Vettraino et al. [46] reported a detection limit of 200 pg for a PCR assay intended to detect *Sclerotinia pseudotuberosa*, another pathogen of chestnut. These techniques, apart from being less sensitive than our mPCR, lack internal controls, which make them prone to false-negative results. On the other hand, Vettraino et al. [29] designed loop-mediated isothermal amplification and real-time PCR assays that were able to detect down to 0.128 pg/μL of *G. smithogilvyi* gDNA. However, these latter techniques have no internal controls and can be considerably more expensive than endpoint-PCR techniques [20], such as the one developed in our study.

Our validation of the mPCR with naturally infected nuts further confirmed the results of previous studies that showed a high incidence of the pathogen in Australian chestnuts. Of the symptomatic and symptomless nuts that we tested, 46% were infected with *G. smithogilvyi*. This result is in agreement with Shuttleworth et al. [7], who found infection levels higher than 30% in nuts sourced from some localities in Victoria, Australia. On the other hand, the detection of *G. smithogilvyi* in symptomless nuts demonstrates the potential of the mPCR as a diagnostic tool. This technique could, for example, be used for monitoring and detecting infections at early stages and even before symptoms appear. Thus, growers would have timely information that would enable them to design and implement effective control strategies for chestnut rot. For example, in our recent report on the use of fungicides to control *G. smithogilvyi* in the field [18], we used this mPCR technique, across a growing season, to monitor the level of nut infection after fungicide applications.

Information about phenotypic differences among *G. smithogilvyi* isolates is limited, and experiments have been carried out using only a few morphotypes. For example, Pasche et al. [47] analysed the differences in conidial length and colony growth rate of two *G. smithogilvyi* isolates, Ge1 and Ti1, sourced from two regions of Switzerland. The authors found that Ti1 had significantly larger conidia than Ge1, but Ge1 displayed a substantially faster growth rate than Ti1. However, due to the limited number of morphotypes analysed, it was difficult to draw any conclusions from the comparison of the phenotypic characteristics of the populations. In contrast, we carried out a larger and more comprehensive phenotypic characterisation of *G. smithogilvyi*, analysing twenty-four isolates sourced from different locations. Among these isolates, phenotypic traits varied significantly between and within populations, especially in colony growth and conidial size. Moreover, we also evaluated the virulence of morphotypes in vitro. While studies in vitro might not be a true reflection of virulence in the field, we have, nevertheless, found significant differences between the isolates tested. For example, isolates from Stanley caused significantly larger lesions in nuts than morphotypes from Bright, Fumina and Wandiligong. Similarly, other studies have also evaluated the differences in virulence of *G. smithogilvyi* morphotypes, although, again, with limited samples. For example, Shuttleworth and Guest [10] measured the lesion length in the nut caused by isolates from Australia (CBS130190), New Zealand (MUT411) and Italy (MUT401). The authors found that the Italian strain MUT401 caused shorter lesions than CBS130190 and MUT411 in nuts of some cultivars. Nevertheless, further studies are still needed to extend our knowledge about the phenotypic variation and its implication for the control of chestnut rot caused by *G. smithogilvyi*.

Finally, we have developed a specific and sensitive multiplex PCR assay to simultaneously detect three barcode sequences of *G. smithogilvyi* and one gene of *C. sativa*. The *C. sativa* gene, *petD*, worked well as an internal control, enhancing the technique’s reliability. Moreover, due to the high sensitivity of this technique, comparable to other more costly assays, the pathogen was able to be detected in visibly symptomless nuts. This technique could, therefore, be used for monitoring and detecting the pathogen at an early stage of infections, providing growers with information for the implementation of timely disease mitigation strategies, such as the use of fungicides. We have also shown that some isolates of *G. smithogilvyi* differed significantly in certain phenotypic traits and that they also varied in their virulence. Understanding the disease pathology and the availability of a sensitive molecular technique to precisely detect the pathogen will assist growers in reducing the impact of *G. smithogilvyi* on chestnut production.

## 4. Materials and Methods

### 4.1. Sample Collection and Fungal Isolation

Symptomatic nuts and burrs of the sweet chestnut (*Castanea sativa* Mill.) were collected from orchards at four locations in Victoria, Australia: Bright (36°41′20.8″ S 147°03′51.0″ E), Fumina (37°54′41.7″ S 146°06′03.7″ E), Stanley (36°24′16.0″ S 146°45′10.3″ E) and Wandiligong (36°47′19.4″ S 146°58′03.8″ E). For the isolation of *G. smithogilvyi* and any other fungal species, samples were surface disinfected in 85% ethanol (1 min), sterile distilled water (1 min), 2.5% NaClO for 10 min, and rinsed with sterile distilled water twice for 2 min. Finally, samples were air-dried on a sterile paper towel in a laminar flow hood. Burrs and nut endosperms were sectioned with a sterile scalpel into 0.5 cm^3^ cubes approximately, and four cubes were placed onto potato dextrose agar medium (PDA, Difco^TM^, Franklin Lakes, NJ, USA). Plates were incubated at 23 °C in the dark for 5 days. To ensure that isolates of all fungi were monospecific, a section from the growing edge of the colony was taken with a sterile cork borer (6 mm) and cultured on fresh PDA medium. Then, plates were incubated, as described above. This procedure was performed twice before carrying out morphological characterisation and DNA extraction.

### 4.2. DNA Extraction and Molecular Identification

Genomic DNA (gDNA) was extracted from approximately 50 mg of fresh fungal tissue with a commercial kit (Quick-DNA^TM^ Fungal/Bacterial Miniprep Kit, Zymo Research) following the manufacturer’s instructions. DNA quality and concentration were determined using a NanoDrop^TM^-1000 (ThermoFisher Scientific, Waltham, MA, USA). Amplification by PCR of the Internal Transcribed Spacer (ITS) region was carried out in a 25 μL reaction volume containing: 1 μL of gDNA (50 ng/μL), 12.5 μL of MyTaqTM HS Mix (Bioline, London, UK), 0.5 μL of each universal primer ITS5/ITS4 (10 μM) [48] and 10.5 μL of nuclease-free water (NFW). Amplification was performed in a master cycler (Nexus gradient, Eppendorf^TM^, Hamburg, Germany) under the following conditions: initial denaturation at 95 °C (2 min), 30 cycles of denaturation at 95 °C (30 s), annealing at 55 °C (30 s), extension at 72 °C (1 min) and a final extension at 72 °C (10 min). PCR products were confirmed by electrophoresis in 1% agarose containing 1 μL/mL of GelRed^®^ (Biotium, Fremont, CA, USA) at 80 Volts for 1 h. PCR products were Sanger sequenced by the Australian Genome Research Facility (Melbourne, VIC, Australia).

### 4.3. Phylogenetic Analysis of G. smithogilvyi Isolates

To confirm the identity of *G. smithogilvyi,* the partial ITS1-5.8S-ITS2 sequences of isolates obtained in this study were compared with 31 sequences of the fungus that were available in the NCBI’s GenBank database (Table 3). These sequences corresponded to isolates from Australia (3), Chile (2), Greece (5), India (5), Italy (4), Portugal (4), Spain (3), and Switzerland (5). All sequences were aligned, edited, and analysed for their phylogenetic relationship and diversity with MEGA (ver 7.0) [49]. The phylogenetic analysis also included three sequences for each of the following *Gnomoniopsis* species, *G. chinensis*, *G. daii*, *G. comari*, *G. fructicola*, and *G. idaeicola*. The sequence of *Penicillium commune* (accession no. NR111143) was used as an outgroup sequence. The consensus phylogenetic tree was constructed based on the maximum likelihood method with 1000 bootstrap replications. The Kimura two-parameter model with a discrete Gamma distribution was selected for nucleotide substitution. Gaps or missing data were treated as partial deletion. The *G. smithogilvyi* ITS sequences obtained in the current study were submitted to the GenBank database under the accession numbers presented in Table 3.

### 4.4. Species-Specific Primer Design

The specific primers for *G. smithogilvyi* were designed with Primer-blast [60] by using the template sequences; translation elongation factor 1α (*TEF*) (accession No. KR072535); Internal Transcribed Spacer (ITS) (accession No. MH865606), and β-tubulin (*TUB*) (accession No. JQ910641). The *C. sativa* gene *petD* was used as the internal control for the mPCR. The amplification of this gene was carried out with the primer pair forward Cmcs1-F 5′-ATTCATTTCCTTTGCATTGA-3′ and reverse Cmcs1-R 5′-TTTACTTGTTACTAATAGGGTCTAGC-3′ [61]. Primers were selected based on their guanine and cytosine content (G-C 40–60%), dinucleotide repetition and melting temperature. Additionally, we evaluated the potential of the primers for forming secondary structures, including heterodimers, self-dimers and hairpins. A Gibbs free energy (ΔG kcal/mol) threshold of −9 kcal/mol was considered the selection cut-off. Primer pairs that potentially formed the secondary structures with energy lower than the threshold were discarded [62]. The online OligoAnalyzer^TM^ tool (Integrated DNA Technologies, Coralville, IA, USA) was used to conduct the analysis [63]. Selected primer pairs were synthesised by Bioneer Pacific (Victoria, Australia).

### 4.5. Multiplex PCR Optimisation

The optimal annealing temperature for each primer pair was determined through a thermal gradient from 51.8 to 59.7 °C in a thermocycler (Eppendorf^TM^). After this, the optimal primer concentration was determined experimentally in a total reaction volume of 20 μL, which included: 1 μL (10 ng/μL) of the pathogen and host gDNA and 10 μL MyTaq^TM^ HS Mix per reaction. Optimal primer concentrations were initially tested at 0.25 μM/primer. Then each primer was tested across a range of concentrations from 0.05 μM/reaction down to 0.5 μM/reaction. When needed, nuclease-free water (NFW) was used to complete the final volume of 20 μL and for negative controls. PCR products were separated and analysed by electrophoresis, as described above.

### 4.6. Multiplex PCR Specificity

The specificity of the multiplex PCR was tested in vitro at the optimised annealing temperature and primer working concentrations. We used 1 μL (10 ng/μL) of pure gDNA from each of the twenty-four *G. smithogilvyi* isolates and from ten of the unrelated fungal species isolated from the nuts. Additionally, two closely related species, *Gnomoniopsis fructicola* (VPRI-41909) and *G. idaeicola* (VPRI-41731), were purchased from the Victorian Plant Pathology Herbarium (Victoria, Australia) and used to evaluate cross-reactivity. All tubes were spiked with 1 μL (10 ng/μL) of the host gDNA as the internal control. Nuclease-free water was used for negative controls. The resulting amplicons (*TEF*, ITS and *TUB*) from three representative *G. smithogilvyi* isolates from each population were sequenced as described above. Sequences were analysed to determine length average (MEGA) and identity in the NCBI-BLAST platform. All sequences had more than 97% identity to *G. smithogilvyi* ex-type culture CBS 130190; *TEF* (KR072534), ITS (NR_166040) and *TUB* (JQ910639) (Appendix A).

### 4.7. Multiplex PCR Detection Limit

*G. smithogilvyi* gDNA was used to determine the detection limit of the mPCR using a dilution series starting from 1000 pg down to 0.1 pg in 20 μL of final reaction volume. We simulated the analytical condition where the host gDNA exceeds the pathogen gDNA by spiking each tube with 1 μL (10 ng/μL) of chestnut gDNA. Nuclease-free water was used for negative controls. Reactions were carried out under the optimised conditions described previously.

### 4.8. Multiplex PCR Validation

Validation of the mPCR was carried out with 50 nuts sourced from Stanley. Nuts were surface-disinfected as described previously and cut open under sterile conditions. Nut tissue (100 mg) was excised with a sterile 3 mm-in-diameter biopsy punch (Kai) and transferred to a 2 mL tube for DNA extraction with a Quick-DNA^TM^ Plant/Seed Miniprep Kit (Zymo Research) following the manufacturer’s instructions. Nuts were categorised as either symptomless or symptomatic with or without pathogen detection by the mPCR. Nut tissues were also cultured on a PDA medium to confirm the presence of *G. smithogilvyi* or other fungal species in the samples. Plates were incubated at 25 °C in the dark for 5 days.

### 4.9. Morphological Characterisation of Colonies

Analysis of colony growth and morphology was performed on PDA. Petri dishes (90 mm) were inoculated with a 6 mm-in-diameter plug of a six-day-old culture of each *G. smithogilvyi* isolate. Plates were sealed with Parafilm^®^ (Amcor, Zürich, Switzerland) and incubated at 25 °C for six days. Mycelial growth was estimated by measuring the colony diameter in two perpendicular directions. The experiment was performed with four replicates per isolate and repeated twice.

### 4.10. Morphological Characterisation of Conidia

For characterising conidial morphology, isolates were induced to sporulate on Cornmeal agar (CMA, Difco^TM^) in Petri dishes (90 mm). Plates were incubated at room temperature with a natural cycle of light and dark for ten days. To enable the consistent and specific isolation of conidia from each isolate, the following procedure was performed: Conidia of each isolate were harvested by adding 5 mL of sterile water and then scraping the colony with a sterile microscope slide. The conidial suspension was force-filtered through a 5 mL pipette tip containing 4-time folded sterile miracloth (Merck, Rahway, NJ, USA) into a 50 mL conical tube by centrifugation for 2 min at 5000 rpm (Eppendorf^TM^ 5804R, Hamburg, Germany). The conidial density was determined using a hemocytometer (Hirschmann, Eberstadt, Germany) and adjusted to 1.0 × 10^6^ conidia/mL. For measuring conidial size, the conidial suspension (10 μL) was placed on a microscope slide, and 30 conidia were randomly selected and assessed using a microscope (Zeiss Axioscope M2, Zeiss, Oberkochen, Germany).

### 4.11. Assessment of G. smithogilvyi Isolate Virulence

Three representative isolates per population were selected to evaluate their virulence in nuts. One hundred and fifty nuts were sourced from an orchard in Stanley; 20 were randomly selected and cut open to confirm the absence of rot symptoms. The remaining 130 nuts were surface disinfected as described previously. Nutshell and pellicle were removed from one side of the nuts with a sterile 3 mm-in-diameter biopsy punch (Kai, Tokyo, Japan). Then, a same-sized PDA plug from each *G. smithogilvyi* isolates (six-day-old culture) was placed in the incision, ensuring that the mycelium was in contact with the nut endosperm. Sterile PDA plugs were used for control treatments. For each isolate, 8 nuts were inoculated and incubated with their respective control nuts at 25 °C for 8 days in the dark. A sterile container arranged with a sterile moist paper towel was used to maintain high humidity during incubation. Nut lesions were recorded with a digital camera (Nikon D5200, Tokyo, Japan), and the lesion area was estimated with Image J (version 1.51j8, U.S. National Institutes of Health, Bethesda, MD, USA).

### 4.12. Statistical Analysis and Data Visualisation

Statistical analysis and graphical visualisation were performed with GraphPad Prism 8 (GraphPad Software, San Diego, CA, USA). Statistical differences in colony and conidia sizes and virulence were determined with one-way ANOVA, and Tukey’s test was used for multiple comparisons at *p* = 0.05. The percentage of nut infection caused by *G. smithogilvyi* in the mPCR validation experiment was compared with a 2 × 2 contingency table and Fisher’s exact test at *p* = 0.05.

## Figures and Tables

**Figure 1 pathogens-11-00907-f001:**
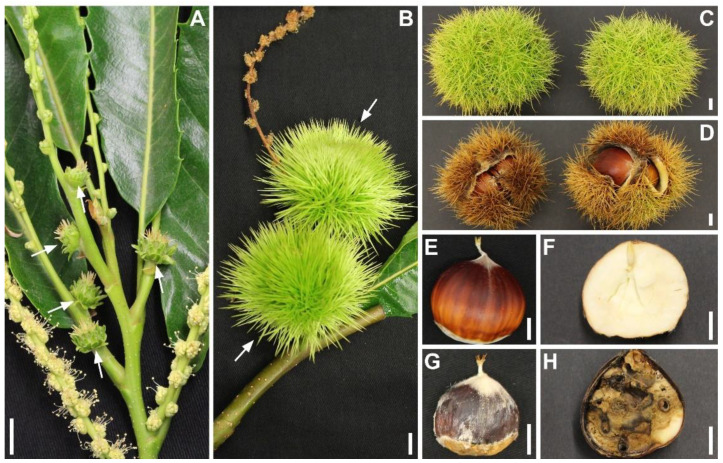
The initial phase of *C. sativa* flowering and infection symptoms caused by *G. smithogilvyi*. (**A**) Immature female flowers (white arrows). Note the bracts in the flower involucre that will become the burr. (**B**) Spiky burrs (white arrows) represent the female flower in development. (**C**,**D**) Symptomless and symptomatic burrs, respectively. (**E**,**F**) External and internal view of symptomless nuts, respectively. (**G**,**H**) External and internal view of symptomatic nuts, respectively. Note the mycelium growth on the external nut surface. Scale bar = 1 cm.

**Figure 2 pathogens-11-00907-f002:**
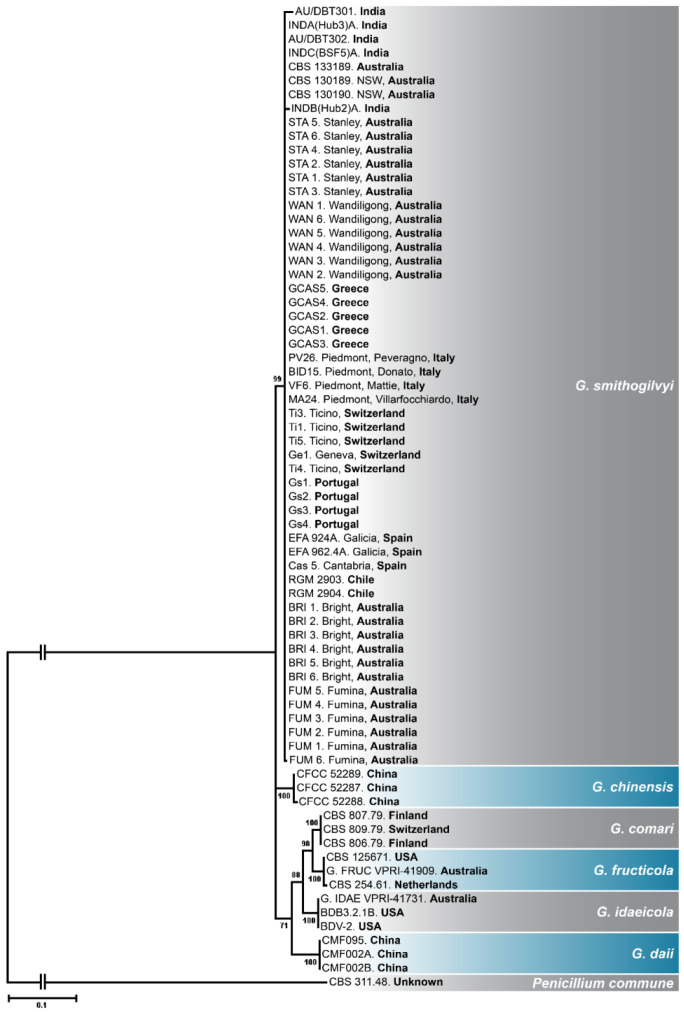
Maximum likelihood phylogenetic tree constructed based on the Kimura 2-parameter model. The ML tree was derived from 1000 bootstrap replications, and the tree with the highest log likelihood (−3084.25) is presented. *Penicillium commune* was used as an outgroup for rooting the tree. The supporting values (>70%) for the taxa clustering are presented on nodes. The scale bar measures the number of substitutions per site.

**Figure 3 pathogens-11-00907-f003:**
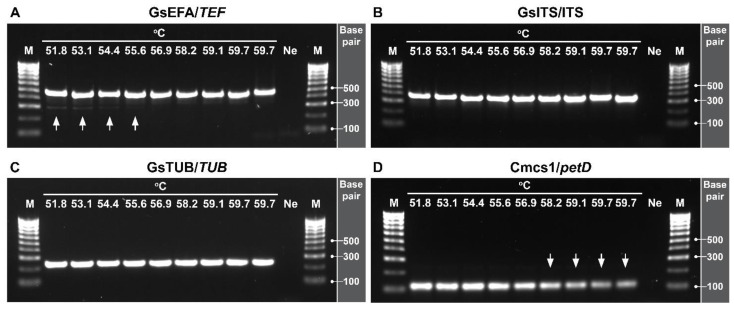
Optimisation of annealing temperature (T_a_) through a thermal gradient from 51.8 to 59.7 °C. (**A**) Primers GsEFA for the translation elongation factor 1α (*TEF*). Note the unintended amplification (white arrows) produced between 51.8 and 55.6 °C. (**B**) Primer GsITS amplifying the internal transcribed spacer (ITS). (**C**) Primers GsTUB for the *TUB* gene. (**D**) Internal control primers Cmcs1 for the host gene *petD*. Note the reduction in intensity (white arrows) produced from 58.2 °C. Optimal T_a_ was established at 56.9 °C. Lane M: molecular marker (100 bp); lane Ne: negative control (NFW).

**Figure 4 pathogens-11-00907-f004:**
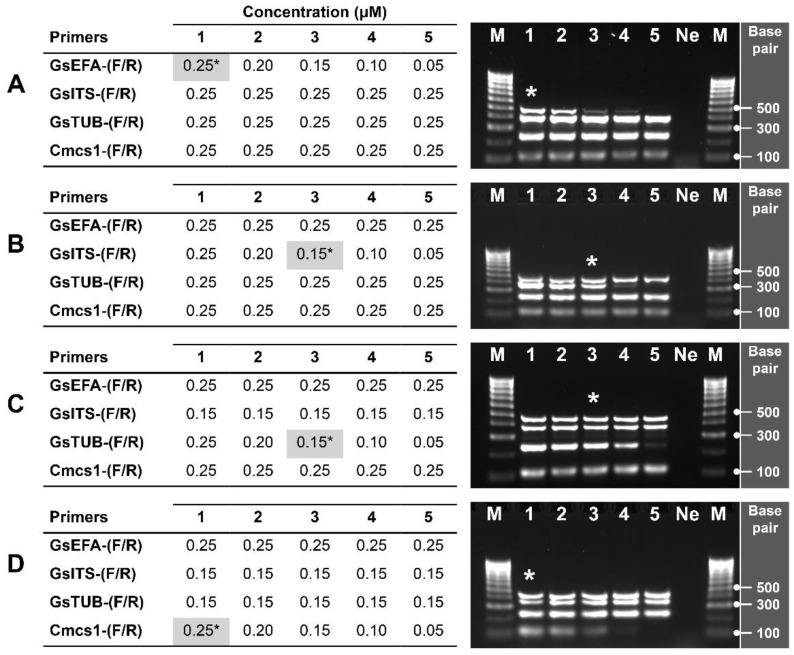
Optimisation of primer concentration for the specific detection of *G. smithogilvyi*. Range of concentrations tested for each primer pair: (**A**) GsEFA, (**B**) GsITS, (**C**) GsTUB and (**D**) the internal control primers Cmcs1. The tables and images show (1–5) the exact concentrations and amplicon bands produced by each primer pair. Asterisks (*) denote the optimal concentration chosen for each primer pair which is highlighted in grey. Lane M: molecular marker (100 bp); lane Ne: negative control (NFW).

**Figure 5 pathogens-11-00907-f005:**
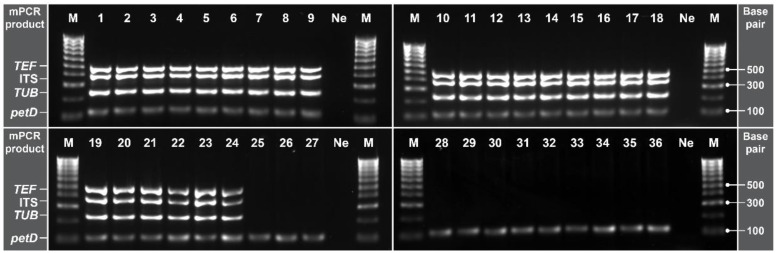
Specificity of the optimised mPCR evaluated with gDNA of *G. smithogilvyi* and other fungal species. Lane M: Molecular marker (100 bp); lane Ne: negative control (NFW). Lanes 1–24: *G. smithogilvyi* isolates. Lane 25, 26 *G. fructicola* and *G. idaeicola,* respectively. Lane 27–36: *Alternaria* sp., *Aspergillus* sp., *Cladosporium* sp., *Clonostachys* sp., *Epicoccum* sp., *Fusarium* sp., *Mucor* sp., *Nigrospora* sp., *Penicillium* sp., and *Phoma* sp., respectively. Note the band of the internal control gene *petD*, which confirmed that the negative results are due to the absence of *G. smithogilvyi* gDNA in the sample and not due to a reaction failure.

**Figure 6 pathogens-11-00907-f006:**
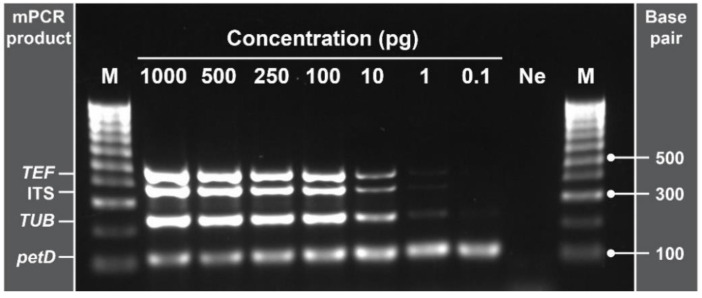
Evaluation of the multiplex PCR detection limit tested with *G. smithogilvyi* gDNA samples spiked with *C. sativa* gDNA. The minimum amount detectable was between 0.1 and 1 pg (5 and 50 fg/μL) of gDNA. Note the consistent band of the internal control gene *petD* at all concentrations. Lane M: molecular marker (100 bp); lane Ne; negative control (NFW). Concentration of 0.1 to 1000 pg represents the amount of *G. smithogilvyi* DNA in 20 μL of the final reaction volume.

**Figure 7 pathogens-11-00907-f007:**
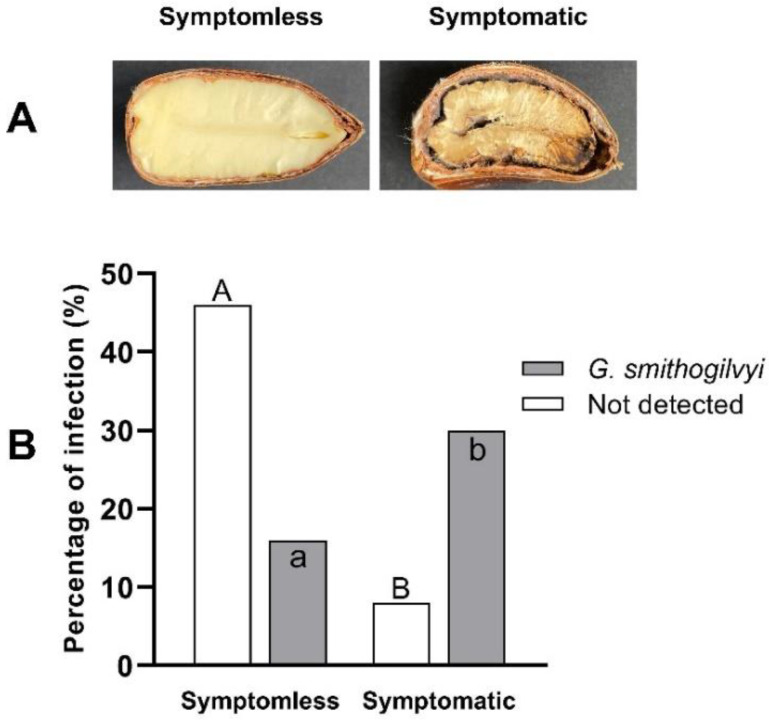
Symptomatology and determination of infection levels by *G. smithogilvyi* through mPCR. (**A**) Representation of symptomless and symptomatic nut. (**B**) Percentage of infection per sample type (*n* = 50). Different uppercase letters for samples in which the pathogen was not detected represent a statistical difference (*p* < 0.0001). Different lowercase letters for samples in which the pathogen was detected represent a statistical difference (*p* < 0.0001). Analysis was based on a 2 × 2 contingency table and Fisher’s test.

**Figure 8 pathogens-11-00907-f008:**
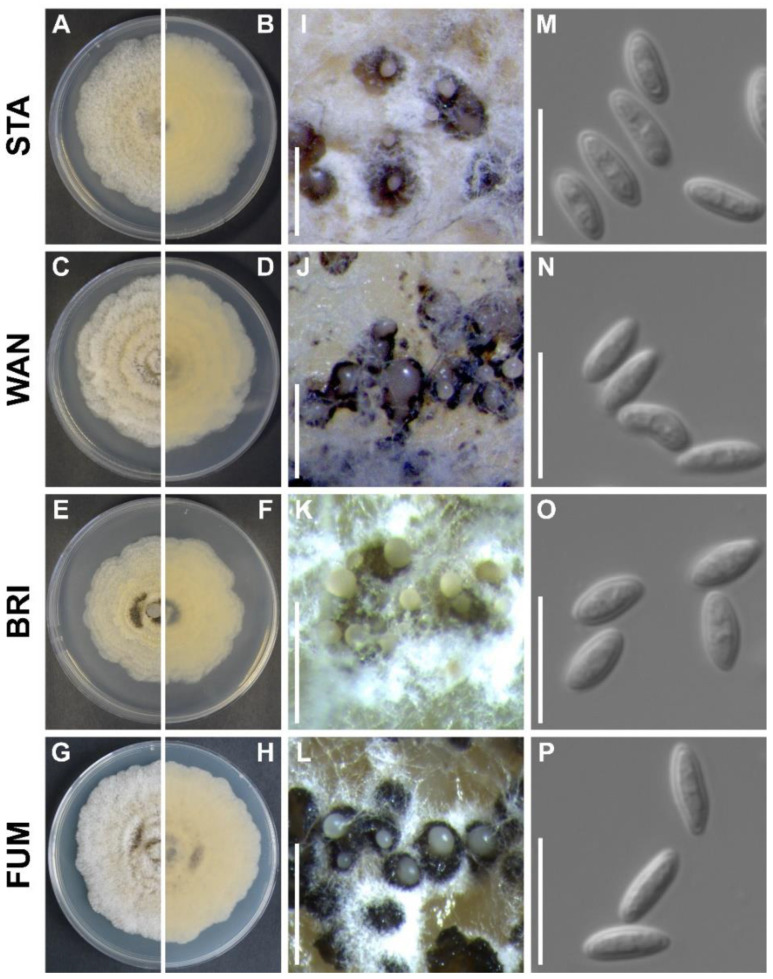
Micro and macromorphological characters of *G. smithogilvyi* representatives from Stanley (STA), Wandiligong (WAN), Bright (BRI), and Fumina (FUM). (**A**–**H**) Upper and reverse colony surfaces. (**I**–**L**) Conidiomata on PDA medium after 10 days at 25 °C. (**M**–**P**) Conidia after growing on CMA medium at room temperature for 10 days. Scale bar: 1 cm (conidiomata), 10 μm (conidia).

**Figure 9 pathogens-11-00907-f009:**
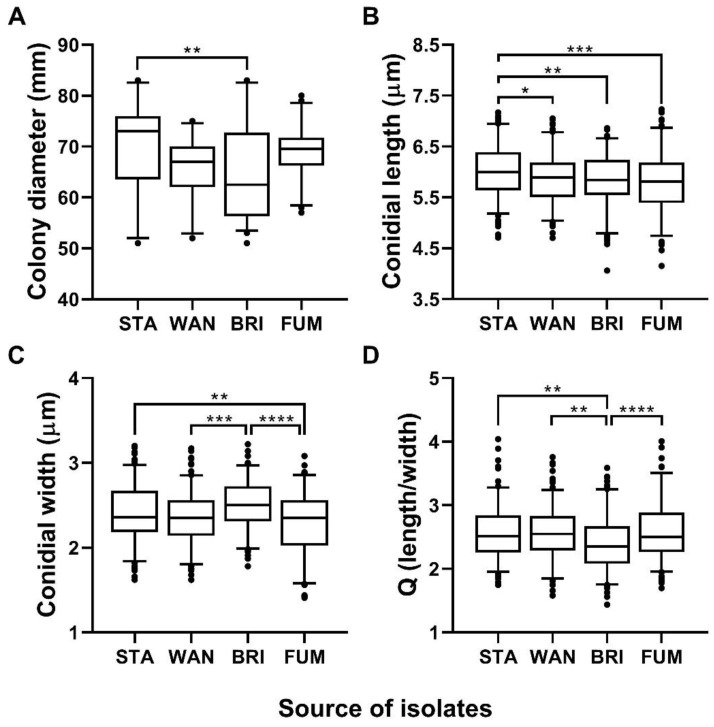
Comparison of colony growth and conidial size means of *G. smithogilvyi* populations from Stanley (STA), Wandiligong (WAN), Bright (BRI) and Fumina (FUM). (**A**) Mean colony growth, (**B**–**D**) mean conidial length, width and Q ratio, respectively. Asterisks represent statistical differences based on Tukey’s test; (*) *p* < 0.05, (**) *p* < 0.01, (***) *p* < 0.001, (****) *p* < 0.0001. Box and whisker plots represent the 5–95 percentile of the sample, and the line within the box is the median.

**Figure 10 pathogens-11-00907-f010:**
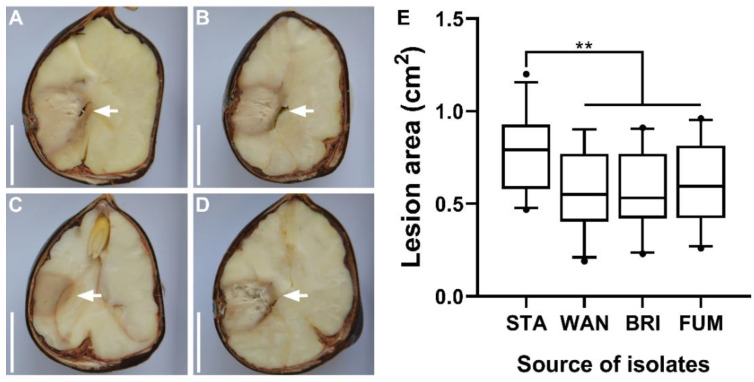
Virulence evaluation of *G. smithogilvyi* representative morphotypes 8 days after inoculation of nuts. Representation of lesions (white arrows) caused by isolates from Stanley (**A**), Wandiligong (**B**), Bright (**C**), and Fumina (**D**). (**E**) Calculated mean lesion based on three morphotypes from each population. Asterisks represent statistical differences based on Tukey’s test; (**) *p* < 0.01). Box and whisker plot represents the 5–95 percentile of the sample, the line within the box is the median.

**Table 1 pathogens-11-00907-t001:** Comparative matrix analysis of ITS sequences derived from *G. smithogilvyi* populations.

Population	Sequence Properties											
	Mean Length (bp)	Site Variability ^a^ (%)		Intrapopulation Divergence (%)	Interpopulation Divergence (%)							
		Variable	Conserved		Australia	Chile	Greece	India	Italy	Portugal	Spain	Switzerland
Australia	568	3.12	84.22	0.02 (± 0.02) ^b^	-	0.01 (± 0.01)	0.01 (± 0.01)	0.43 (± 0.13)	1.02 (± 0.45)	0.01 (± 0.01)	0.35 (± 0.15)	0.01 (± 0.01)
Chile	611	0.14	90.47	0.00 (± 0.00)	-	-	0.00 (± 0.00)	0.43 (± 0.13)	1.06 (± 0.47)	0.00 (± 0.00)	0.35 (± 0.16)	0.00 (± 0.00)
Greece	539	0	80.20	0.00 (± 0.00)	-	-	-	0.43 (± 0.13)	1.06 (± 0.47)	0.00 (± 0.00)	0.35 (± 0.16)	0.00 (± 0.00)
India	554	4.01	80.35	0.78 (± 0.23)	-	-	-	-	1.08 (± 0.39)	0.43 (± 0.13)	0.58 (± 0.18)	0.43 (± 0.13)
Italy	559	0.14	84.07	0.00 (± 0.00)	-	-	-	-	-	1.06 (± 0.47)	0.35 (± 0.16)	1.06 (± 0.47)
Portugal	567	0	84.37	0.00 (± 0.00)	-	-	-	-	-	-	0.35 (± 0.16)	0.00 (± 0.00)
Spain	535	0.15	76.04	0.35 (± 0.15)	-	-	-	-	-	-	-	0.35 (± 0.16)
Switzerland	530	0	79.01	0.00 (± 0.00)	-	-	-	-	-	-	-	-

^a^ Site variability was calculated as (%) based on the number of sites per population divided by the total number of aligned sites. ^b^ Values in brackets for intra- and interpopulation divergence represent the standard error of the mean.

**Table 2 pathogens-11-00907-t002:** Sequences and parameters of the selected species-specific primers for detection of *G. smithogilvyi* through mPCR.

Species	Target Gene	Primer Name	Primer Sequence 5′ to 3′	GC (%)	T_m_ ^d^ (°C)	ΔG (kcal/mol) ^e^				Product Length (bp) ^f^	Reference
						Harpin	Self	Hetero			
								(a)	(b)		
*G. smithogilvyi*	*TEF* ^a^	GsEFA-F	TCTTCATCGTCGATTCCTTG	45	52.1	1.1	−6.76	−6.59	−8.26 – −3.55	483	This study
		GsEFA-R	GAGCTGTGGAACCAACACCAA	52	57.9	−0.6	−6.34				
	ITS ^b^	GsITS-F	GGCTTCCTATGGAAGTCCCTC	57	57.0	−2.83	−8.19	−6.21		367	This study
		GsITS-R	CAAGAGCAACCGCCAGTCTT	55	58.0	−0.4	−5.12				
	*TUB* ^c^	GsTUB-F	ATCAACCCCTTCAGAGACGC	55	57.1	0.13	−3.61	−7.07		203	This study
		GsTUB-R	ACGTGAAGCTCAAGTACGCA	50	56.8	−0.96	−6.34				

^a^ Translation Elongation Factor 1α; ^b^ Internal Transcribed Spacer; ^c^ β-tubulin; ^d^ Melting temperature calculated with OligoAnalyzer^TM^. ^e^ Gibbs free energy (ΔG kcal/mol) for Harpins, Self-dimers (Self) and Hetero-dimers (Hetero): among primer pair (a) and range between all primers (b). ^f^ mPCR product length based on Appendix A.

**Table 3 pathogens-11-00907-t003:** List of *G. smithogilvyi* and other fungal species ITS sequences used in this study.

Species	Isolate ID	Plant Host	Locality	ITS GenBank ID	Reference
*G. smithogilvyi*	BRI 1	*C. sativa*	Bright, Australia	ON545732	This study
	BRI 6			ON545733	
	BRI 3			ON545734	
	BRI 4			ON545735	
	BRI 5			ON545736	
	BRI 6			ON545737	
	FUM 1	*C. sativa*	Fumina, Australia	ON545744	
	FUM 2			ON545745	
	FUM 3			ON545746	
	FUM 4			ON545747	
	FUM 5			ON545748	
	FUM 6			ON545749	
	STA 1	*C. sativa*	Stanley, Australia	ON545750	
	STA 2			ON545751	
	STA 3			ON545752	
	STA 4			ON545753	
	STA 5			ON545754	
	STA 6			ON545755	
	WAN 1	*C. sativa*	Wandiligong, Australia	ON545738	
	WAN 2			ON545739	
	WAN 3			ON545740	
	WAN 4			ON545741	
	WAN 5			ON545742	
	WAN 6			ON545743	
*G. smithogilvyi*	CBS 133189	*Castanea* sp.	Australia	KY952223	[10]
	CBS 130189			MH865606	[50]
	CBS 130190			NR_166040	
	RGM 2903	*C. sativa*	Chile	MT413428	[12]
	RGM 2904			MT413429	
	AU/DBT301	*C. sativa*	India	KC963935	[14]
	AU/DBT302			KC963936	
	INDA(Hub3)A			JQ268071	
	INDB(Hub2)A			JQ268072	
	INDC(BSF5)A			JQ268073	
	Gs1	*C. sativa*	Portugal	MW165483	[51]
	Gs2			MW165484	
	Gs3			MW165485	
	Gs4			MW165486	
*G. smithogilvyi*	Cas 5	*C. sativa*	Cantabria, Spain	KU095876	[52]
	EFA 962.4A		Galicia, Spain	OM319848	[16]
	EFA 924A			OM319846	
	Ge1	*C. sativa*	Geneva, Switzerland	KP824754	[47]
	Ti1		Ticino, Switzerland	KP824746	
	Ti3			KP824748	
	Ti4			KP824750	
	Ti5			KP824752	
*G. castaneae* ^a^	BID15	*C. sativa*	Donato, Piedmont, Italy	LN999963	[53]
	MA24		Mattie, Piedmont, Italy	LN999969	
	PV26		Peveragno, Piedmont, Italy	LN999967	
	VF6		Villarfocchiardo Piedmont, Italy	LN999964	
	GCAS1	*C. sativa*	Greece	MH107826	[13]
	GCAS2			MH107827	
	GCAS3			MH107828	
	GCAS4			MH107829	
	GCAS5			MH107830	
*G. chinensis*	CFCC 52287	*C. mollissima*	China	MG866033	[54]
	CFCC 52288			MG866034	
	CFCC 52289			MG866035	
*G. comari*	CBS 806.79	*Comarum palustre*	Finland	EU254821	[55]
	CBS 807.79		Finland	EU254822	
	CBS 809.79		Switzerland	EU254823	
*G. daii*	CMF002A	*C. mollissima*	China	MN598671	[56]
	CMF002B			MN598672	
	CMF095			MN598673	
*G. fructicola*	G_FRUC_VPRI-41909	*Fragaria* sp.	Australia	ON545716	This study
	CBS 125671	Unknown	USA	MH863616	[50]
	CBS 254.61			MH858043	
*G. idaeicola*	BDB3.2.1B	*Rubus* sp.	USA	OK348854	[57]
	BDV-2			OK348857	
	G_IDAE_VPRI-41731	*R. fruticosus*	Australia	ON545717	This study
*Mucor* sp.	Iso1	*C. sativa*	Australia	ON545707	This study
*Penicillium* sp.	Iso4			ON545708	
*Clonostachys* sp.	Iso8			ON545718	
*Epicoccum* sp.	Iso14			ON545711	
*Nigrospora* sp.	Iso15			ON545710	
*Alternaria* sp.	Iso25			ON545709	
*Fusarium* sp.	Iso26			ON545712	
*Phoma* sp.	Iso27			ON545713	
*Cladosporium* sp.	WN13			ON545714	
*Aspergillus* sp.	WM9			ON545715	
*P. commune*	CBS 311.48	Unknown	Unknown	NR111143	[58]

^a^  *G. castaneae* and *G. smithogilvyi* are synonyms of the same plant pathogen [59].

## Data Availability

The datasets generated during and/or analysed during the current study are not publicly available due to commercial confidentiality but are available from the corresponding author on reasonable request.

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
