# Peer review of "Rapid and Accurate Detection of Gnomoniopsis smithogilvyi the Causal Agent of Chestnut Rot, through an Internally Controlled Multiplex PCR Assay"

_pathogens, 2022, doi:10.3390/pathogens11080907_

Round 1
Reviewer 1 Report
The manuscript entitled “Rapid and Accurate Detection of Gnomoniopsis smithogilvyi the Causal Agent of Chestnut Rot, through an Internally Controlled Multiplex PCR Assay” by Matias Silva-Campos and colleagues reports a useful method for detecting a phytopathogenic fungus caused chestnut rot disease. The manuscript is composed of two parts: 1) development of a detection method based on multiplex PCR. 2) characterization of G. smithogilvyi isolates varied in morphology and virulence. The manuscript is well written in a straightforward and succinct way with very solid data on the detection of phytopathogenic fungi. I recommend this paper to be accepted for publication in Pathogens after appropriate minor revisions.
These are some comments for further improvement:
Line 100. Table 1 is better provided as a supplementary table.
Line 120. “The comparison of the sequence alignment …” is not clear and should add the name of gene sequence.
Line 145. The gene symbols should be italicized. Please go through the entire manuscript and modify it.
Line 185. Figure 3-6. What is the “negative control”? Should define it.
Line 264. Figure 7. It is difficult to understand figure 7B. I suggest separating the percentage of infection of G.smithogilvyi and un- G.smithogilvyi isolates into two columns.
The format of reference should be unified.
Author Response
Dear reviewer,
The authors of this manuscript are grateful for all the valuable, constructive comments and suggestions received. The comments have been of great help and useful to improve our manuscript. We have revised and included the comments in the last version of our manuscript in the Word file. Please keep in mind that the suggested page line numbers correspond to the manuscript version with Track changes deactivated and No markup. Our responses to your comments and suggestions are as follow:
Comment 1: Line 100. Table 1 is better provided as a supplementary table.
Response 1: Thank you for this suggestion. We would prefer to leave the table within the body of the manuscript. Table 1 has therefore been moved to the Material and Methods section, following the journal guidelines regarding accession numbers. Note that in the revised version of the manuscript Table 1 has become Table 3 (page 17-19).
Comment 2: Line 121. “The comparison of the sequence alignment …” is not clear and should add the name of gene sequence.
Response 2: Thank you for the suggestion. We have included the name of the gene ITS in the sentence. Line 121
Comment 3: Line 145. The gene symbols should be italicized. Please go through the entire manuscript and modify it.
Response 3: Thank you for pointing this out. We have italicised all gene symbols through the manuscript, figures and tables.
Comment 4: Line 185. Figure 3-6. What is the “negative control”? Should define it.
Response 4: The negative control was nuclease free water (NFW) and is now stated in the legend in Figures 3-6. Additionally, we have defined NFW in the Material and Methods sections, lines 491 and 540. We have also specified the use of nuclease free water as the negative control in lines 552 and 562.
Comment 5: Line 264. Figure 7. It is difficult to understand figure 7B. I suggest separating the percentage of infection of G.smithogilvyi and un- G.smithogilvyi isolates into two columns.
Response 5: We have modified Figure 7B as suggested. Now the bars in Figure 7 have been separated and figure legend updated accordingly.
Comment 6: The format of reference should be unified.
Response 6: We believe the reference format is unified throughout the manuscript and the reference list. We used Pathogen-MDPI reference style in EndNote for this purpose.
Reviewer 2 Report
General comments
The paper is generally well written and has a good structure and flow. The science is sound in general. I would suggest some small changes to improve the paper and notes are provided as to what these suggestions are. Good methods and controls for the DNA work.
Some sentences do not follow well and could be reviewed. These are;
Lines 41 to 43 one or more commas would help with the word follow
lines 44 to 46 a comma pathogenic
line 47 a comma after disease
Multiple times in the paper the phrase disease management strategies is used with no clear reference to what these might be and how they would be effective in controlling the pathogen. Either list controls such as rapidly drying nuts or other controls that are currently used or suggest that the test method provides the industry information that targets the diseases cycle for new control techniques and a method to test eh effectiveness of these when implemented. The relative expense of the test in the discussion is only important if growers will use it to control the disease.
Line 323 to 345 provides information on the relative growth rates of the pathogen on the nut substrate under artificial conditions. I feel it is a stretch to call this virulence as an agar plug of pure culture is very different to field infection that would normally have to penetrate the seed coat and other physical deference. Virulence is more than just the ability to grow or rate of growth once infection is successful. Discussion of the importance of these growth measurements therefore should be toned down in the discussion.
Author Response
Dear reviewer,
The authors of this manuscript are grateful for all the valuable, constructive comments and suggestions received. The comments have been of great help and useful to improve our manuscript. We have revised and included the comments in the last version of our manuscript in the Word file. Please keep in mind that the suggested page line numbers correspond to the manuscript version with Track changes deactivated and No markup. Our responses to your comments and suggestions are as follow:
Comment 1: Lines 41 to 43 one or more commas would help with the word follow
Response 1: Thank you for the suggestion. To improve the word flow we have removed the name of the authorities for G. smithogilvyi as suggested by another reviewer. Please refer to lines 41 – 42.
Comment 2: lines 44 to 46 a comma pathogenic
Response 2: We have included a comma after pathogenic. Line 44
Comment 3: line 46 a comma after disease
Response 3: We have included a comma after disease. Line 46
Comment 4: Multiple times in the paper the phrase disease management strategies is used with no clear reference to what these might be and how they would be effective in controlling the pathogen. Either list controls such as rapidly drying nuts or other controls that are currently used or suggest that the test method provides the industry information that targets the diseases cycle for new control techniques and a method to test eh effectiveness of these when implemented. The relative expense of the test in the discussion is only important if growers will use it to control the disease.
Response 4: We thank the reviewer for this suggestion. We have included a statement in the Introduction (line 54) indicating the use of fungicides as a way of controlling diseases that depend on the early detection of the pathogen. In addition to this, we included in the Discussion an example and reference of how our mPCR has been used for monitoring G. smithogilvyi after fungicide applications in the field. Lines 427-430.
Comment 5: Line 323 to 345 provides information on the relative growth rates of the pathogen on the nut substrate under artificial conditions. I feel it is a stretch to call this virulence as an agar plug of pure culture is very different to field infection that would normally have to penetrate the seed coat and other physical deference. Virulence is more than just the ability to grow or rate of growth once infection is successful. Discussion of the importance of these growth measurements therefore should be toned down in the discussion.
Response 5: We agree with the reviewer, and we have modified the Results section stating that the virulence assay was carried out in vitro (lines 330-331). In addition to this, we restated in the Discussion section to make clear that our assay was performed under in vitro conditions. Lines 442-444
Reviewer 3 Report
- The manuscript entitled “Rapid and Accurate Detection of Gnomoniopsis smithogilvyi, the Causal Agent of Chestnut Rot, through an Internally Controlled Multiplex PCR Assay is written in very casual and without strong proof reading. The concept is fine but the study requires further validation using qPCR assay to verify the results of multiplex assay, most importantly, melting curve analysis. In present form manuscript seems too week at both technical and limited number of isolates selected for the development and validation of assay.
- The information provided in the manuscript is not presented in well manner. It is beyond my understanding why authors included the characterization of limited number isolates based on morphology and virulence? It will be better if author’s first describe the characterization the Gnomoniopsis smithogilvyi isolates based on morphology, followed by their molecular identification, virulence analysis of isolates and at last diagnostic assay development and its validation information.
- On what basis authors selected Tef1a, ITS and ß-tubulin regions to develop primers. Include this information in the introduction with suitable references. Why authors not used all the three genes to establish the molecular identity as ITS alone is not sufficient to confirm the isolate identity at genomic level.
- It is unclear how many isolates authors used in the validation of diagnostic assay. 55 or something else? 28 (retrieve sequences from databank) + 24 (tested isolates in present study) = 42 ? Check and confirm.
- Line 88-92, the NCBI accession number not matched with the fungal species. Provide complete names of all the fungal species with codes.
- The number of isolates used to check the cross-amplification is very low. Authors should use at least 50- or more fungal isolates to check the cross amplification of the developed markers. It will better if authors include all the fungal pathogen inciting disease in chest nut host in addition to validate the newly developed primers.
- I am not convincing with multiplex PCR detection limit experimentation? Authors should use perform real time PCR machine to confirm or validate the results.
- Gel images are not of good quality and not accurate. The amplicon size of Tef gene should be 518 bp but in Fig 3 it reflected below 500bp. (Fig 3). Similarly, with the case of ITS where its amplicon size mentioned below 400 bp and differs from the information mentioned in Table 3 as 404 bp.
- Were authors also observed cross amplification or primer-dimer formation during optimization of primer concentration? Authors should discuss this part too.
- How the developed assay is better in terms of precision, time and sensitivity than the other earlier reported multiplex PCR based assay? Authors should also add this information in discussion. Below are few references which will be helpful for authors to improve the manuscript: https://doi.org/10.1038/s41598-020-79117-0; https://doi.org/10.3389/fpls.2020.01039; 10.3390/biology10121295;
Author Response
Dear reviewer,
The authors of this manuscript are grateful for all the valuable, constructive comments and suggestions received. The comments have been of great help and useful to improve our manuscript. We have revised and included the comments in the last version of our manuscript in the Word file. Please keep in mind that the suggested page line numbers correspond to the manuscript version with Track changes deactivated and No markup. Our responses to your comments and suggestions are as follow:
Comment 1: The manuscript entitled “Rapid and Accurate Detection of Gnomoniopsis smithogilvyi, the Causal Agent of Chestnut Rot, through an Internally Controlled Multiplex PCR Assay is written in very casual and without strong proof reading.
Response 1: We have improved the manuscript by following the suggestions that each of the four reviewers have provided. We believe that the manuscript is now better constructed and is well written.
Comment 2: The concept is fine but the study requires further validation using qPCR assay to verify the results of multiplex assay, most importantly, melting curve analysis.
Response 2: Thank you for assessing the concept of the manuscript as fine. We have considered your comment around validation using qPCR and melting curve analysis, but we believe that this validation is not necessary. Endpoint PCR assays such as the one developed in our study are widely used in plant pathology as robust and reliable diagnostic techniques themselves without the requirement of further validation. Moreover, one of the critical aspects of our study has been the inclusion of an internal control which is often not considered in other techniques.
Comment 3: In present form manuscript seems too week at both technical and limited number of isolates selected for the development and validation of assay.
Response 3: Thank you for the comment, but we do not quite understand what you mean by too week. We have now made significant changes to the manuscript to strengthen the technical side and to provide an explanation around the number of isolates used.
Comment 4: The information provided in the manuscript is not presented in well manner. It is beyond my understanding why authors included the characterization of limited number isolates based on morphology and virulence? It will be better if author’s first describe the characterization the Gnomoniopsis smithogilvyi isolates based on morphology, followed by their molecular identification, virulence analysis of isolates and at last diagnostic assay development and its validation information.
Response 4: Thank you for the comment. However, we believe that including a phenotypic characterisation of G. smithogilvyi isolates makes our manuscript more comprehensive in the context of opening up our research, to all readers, on the pathogen that causes chestnut rot. Regarding the order of the manuscript, it makes better sense to leave it as is, because the primary focus of this research is the development of an mPCR assay which is then supported by subsequent information around phenotypic characterisation.
Comment 5: On what basis authors selected Tef1a, ITS and ß-tubulin regions to develop primers. Include this information in the introduction with suitable references. Why authors not used all the three genes to establish the molecular identity as ITS alone is not sufficient to confirm the isolate identity at genomic level.
Response 5: Thank you for the comment. We agree with the suggestion, and we have added a sentence justifying the use of the three genes in the Introduction (Lines 66-69). Additionally, we included in Material and Methods and supplementary information the identity of the amplicons (TEF, ITS and TUB) obtained with the specific primers which further support the identification of G. smithogilvyi. Lines 552- 557 and Table S1.
Comment 6: It is unclear how many isolates authors used in the validation of diagnostic assay. 55 or something else? 28 (retrieve sequences from databank) + 24 (tested isolates in present study) = 42 ? Check and confirm.
Response 6: Thank you for the comment. In section 2.5 multiplex PCR specificity we have listed the number and names of isolates used. Twenty-four Gnomoniopsis smithogilvyi isolates, two related species G. fructicola and G. idaeicola, and 10 unrelated species were used for validation. Lines 210 -216 and 546-549.
Comment 7: Line 88-92, the NCBI accession number not matched with the fungal species. Provide complete names of all the fungal species with codes.
Response 7: Thanks for pointing this out. We now provide the full name for each species and the relevant accession number. Lines 91-94
Comment 8: The number of isolates used to check the cross-amplification is very low. Authors should use at least 50- or more fungal isolates to check the cross amplification of the developed markers. It will better if authors include all the fungal pathogen inciting disease in chest nut host in addition to validate the newly developed primers.
Response 8: Thank you for the comment. However, as far as we are aware there is not a minimum number of isolates for PCR validation. Indeed, recent published reports have used less than the 50-threshold suggested. The isolates that we have chosen were those most commonly found in association with nuts. We believe that our total of 36 isolates is sufficient to show specificity in the context of chestnut.
Comment 9: I am not convincing with multiplex PCR detection limit experimentation? Authors should use perform real time PCR machine to confirm or validate the results.
Response 9: Thank you for the comment, we used a very common approach for determining the detection limit of our assay, that is, running a serial dilution of DNA and assessment of the presence or absence of a band after amplification. Regarding the second part of the question, we have also addressed this comment in our response to comment 2 above.
Comment 10: Gel images are not of good quality and not accurate. The amplicon size of Tef gene should be 518 bp but in Fig 3 it reflected below 500bp. (Fig 3). Similarly, with the case of ITS where its amplicon size mentioned below 400 bp and differs from the information mentioned in Table 3 as 404 bp.
Response 10: Thank you for pointing this out. We have now included the correct amplicon sizes in Table 2. The amplicon sizes were determined using the sequence lengths that are now shown in a new supplementary table (Table S1)
Comment 11: Were authors also observed cross amplification or primer-dimer formation during optimization of primer concentration? Authors should discuss this part too.
Response 11: Thank you for the question. We did not observe cross amplification or primer-dimer formation during the optimisation of primer concentration. We have stated this in the Discussion. Lines 399-400.
Comment 12: How the developed assay is better in terms of precision, time and sensitivity than the other earlier reported multiplex PCR based assay? Authors should also add this information in discussion. Below are few references which will be helpful for authors to improve the manuscript: https://doi.org/10.1038/s41598-020-79117-0; https://doi.org/10.3389/fpls.2020.01039; 10.3390/biology10121295;
Response 12: Thank you for the comment. We have discussed how our mPCR is better in terms of sensitivity and reliability than other mPCR techniques used with fungal plant pathogens. Lines 401-405. Thank you as well for suggesting some references, we have included two of them in our Introduction. Line 57
Reviewer 4 Report
This is a well-written manuscript and provides important information on G. smithogilvyi associated with chestnut rot.
I have no major issues with the submitted manuscript and recommend it accepted for publication. However, I suggest the authors consider the comments listed below prior to final acceptance.
Line 12: replace ''nuts from sweet chestnut (Castanea sativa)'' with ''sweet chestnut (Castanea sativa) nuts''.
Line 14: ''leads to'' instead of ''translates into''.
Line 16: ''based on the amplification of''' instead of ''based amplifying''.
Line 18: Use italics for in vitro and in silico throughout the manuscript.
Line 32: Delete Mill.
Line 41: Delete authorities with Latin binomials.
Line 42: castaneae instead of castanea.
Line 43: Transfer ''in Australia'' at the end of the sentence.
Line 68: ''understanding of''.
Table 1: ''Reference'' instead of ''Reference(s)''.
Table 1: Species ''G. castaneae'' instead of ''G. casteaneaea'' (Greece).
Lines 349-350: Delete this sentence. This has already been mentioned in Introduction Section.
Line 452: ''collected'' instead of ''sourced''.
Line 454: ''surface disinfected'' instead of ''surface-sterilized''.
Lines 461-646: Did the authors follow this method to obtain single-spore cultures of their isolates?
Line 484: Add ''Table 1'' after ''GenBank database''.
Lines 483-485: Rephrase these two sentences into one.
Line 486: ''corresponded to isolates'' instead of ''included''.
Line 496: Add at the end of the sentence ''under the Accession Nos. presented in Table 1''.
Line 530: ''tubes'' instead of ''tube''.
Line 549: ''on PDA'' instead of ''on the PDA medium''.
Line 551: G. smithogilvyi in italics.
Line 552: Rephrase to ''was estimated by measuring the colony diameter in two perpendicular directions''.
Lines 560-568: Was all this procedure necessary in order to carry out the morphological characterization of the conidia?
Author Response
Dear reviewer,
The authors of this manuscript are grateful for all the valuable, constructive comments and suggestions received. The comments have been of great help and useful to improve our manuscript. We have revised and included the comments in the last version of our manuscript in the Word file. Please keep in mind that the suggested page line numbers correspond to the manuscript version with Track changes deactivated and No markup. Our responses to your comments and suggestions are as follow:
Comment 1: Line 12: replace ''nuts from sweet chestnut (Castanea sativa)'' with ''sweet chestnut (Castanea sativa) nuts''.
Response 1: We have restated the sentence as suggested. Lines 12-13.
Comment 2: Line 14: ''leads to'' instead of ''translates into''.
Response 2: We have changed the wording as suggested. Line 14.
Comment 3: Line 16: ''based on the amplification of''' instead of ''based amplifying''.
Response 3: We have changed the wording as suggested. Line 16.
Comment 4: Line 18: Use italics for in vitro and in silico throughout the manuscript.
Response 4: We have italicised all the in vitro and in silico words throughout the manuscript as suggested.
Comment 5: Line 32: Delete Mill.
Response 5: We have deleted Mill in line 32 as suggested.
Comment 6: Line 41: Delete authorities with Latin binomials.
Response 6: We have deleted the authorities as suggested. Line 41
Comment 7: Line 42: castaneae instead of castanea.
Response 7: We have changed the epithet to castaneae as suggested. Line 41
Comment 8: Line 43: Transfer ''in Australia'' at the end of the sentence.
Response 8: We have reworded the sentence as suggested. Line 42
Comment 9: Line 68: ''understanding of''.
Response 9: We have reworded the sentence as suggested. Lines 70-71
Comment 10: Table 1: ''Reference'' instead of ''Reference(s)''.
Response 10: We have deleted (s) as suggested. Please be aware that Table 1 has been moved to the Material and Methods section, following the journal guidelines. Table 1 is now Table 3 (pages 17-19)
Comment 11: Table 1: Species ''G. castaneae'' instead of ''G. casteaneaea'' (Greece).
Response 11: We have changed the epithet to castaneae as suggested
Comment 12: Lines 349-350: Delete this sentence. This has already been mentioned in Introduction Section.
Response 12: Thank you for this suggestion. We believe that it is important to restate the relevance of the pathogen for the chestnut industry at the beginning of the Discussion. Nevertheless, we have combined the first two sentences to make the introductory lines more concise. Lines 354-355
Comment 13: Line 452: ''collected'' instead of ''sourced''.
Response 13: We have changed the sentence as suggested. Line 470.
Comment 14: Line 454: ''surface disinfected'' instead of ''surface-sterilized''.
Response 14: We have reworded the sentence as suggested. Line 473
Comment 15: Lines 461-646: Did the authors follow this method to obtain single-spore cultures of their isolates?
Response 15: Thank you for showing interest in our procedure. We obtained pure cultures by subculturing a plug from the growing edge of the colonies instead of single-spore cultures.
Comment 16: Line 484: Add ''Table 1'' after ''GenBank database''.
Response 16: We have made the changes as suggested. Please be aware that Table 1 is now Table 3. Line 515.
Comment 17: Lines 483-485: Rephrase these two sentences into one.
Response 17: We have combined the sentences as suggested. Lines 499-500
Comment 18: Line 486: ''corresponded to isolates'' instead of ''included''.
Response 18: We have made the changes as suggested. Line 501
Comment 19: Line 496: Add at the end of the sentence ''under the Accession Nos. presented in Table 1''.
Response 19: We have reworded the sentence as suggested. Please be aware that Table 1 is now Table 3. Lines 511-512.
Comment 20: Line 530: ''tubes'' instead of ''tube''.
Response 20: We have changed the word as suggested. Line 551
Comment 21: Line 549: ''on PDA'' instead of ''on the PDA medium''.
Response 21: We have reworded the sentence as suggested. Line 576
Comment 22: Line 551: G. smithogilvyi in italics.
Response 22: We have italicized the scientific name as suggested. Line 577
Comment 23: Line 552: Rephrase to ''was estimated by measuring the colony diameter in two perpendicular directions''.
Response 23: Thank you for this suggestion. We have rephrased the sentence as suggested. Lines 579-581
Comment 24: Lines 560-568: Was all this procedure necessary in order to carry out the morphological characterization of the conidia?
Response 24: Thank you for showing interest in our procedure. Yes, in order to enable consistent and specific isolation of conidia from the agar medium, we performed this protocol. We found that filtration is necessary to remove hyphae and agar residues before microscopy and image acquisition. Moreover, the journal requires us to provide as much detail as possible in the Material and Methods section. We have added a statement regarding this. Lines 585-586.
Round 2
Reviewer 3 Report
The authors revised the the last version of our manuscript and also attended also the comments and queries raised by me in last version of the manuscript.